

# Fire intensity impacts on post-fire temperate coniferous forest net primary productivity

Aaron M. Sparks[1], Crystal A. Kolden[1], Alistair M.S. Smith[1], Luigi Boschetti[1], Daniel M. Johnson[1], Mark A. Cochrane[2]

[1]College of Natural Resources, University of Idaho, Moscow, ID, USA
[2] Appalachian Laboratory, University of Maryland, Frostburg, MD, USA

*Correspondence to*: Aaron M. Sparks (spar5010@vandals.uidaho.edu)

**Abstract.** Fire is a dynamic ecological process in forests and impacts the carbon (C) cycle through direct combustion emissions, tree mortality, and by impairing the ability of surviving trees to sequester carbon. While studies on young trees
have demonstrated that fire intensity is a determinant of post-fire net primary productivity, wildland fires at landscape to regional scales have largely been assumed to either cause tree mortality, or conversely, cause no physiological impact, ignoring the impacted but surviving trees. Our objective was to understand how fire intensity affects post-fire net primary productivity in conifer-dominated forested ecosystems at the spatial scale of large wildland fires. We examined the relationships between fire radiative power (FRP), its temporal integral (fire radiative energy - FRE), and net primary productivity (NPP) using 16
years of data from the MOderate Resolution Imaging Spectrometer (MODIS) for 15 large fires in western United States coniferous forests. The greatest NPP post-fire loss occurred one year post-fire and ranged from -67 to -312 g C m$^{-2}$ yr$^{-1}$ (-13 to -54%) across all fires. Forests dominated by fire-resistant species (species that typically survive low intensity fires) experienced the lowest relative NPP reductions compared to forests with less resistant species. Post-fire NPP in forests that were dominated by fire-susceptible species were not as sensitive to FRP or FRE, indicating that NPP in these forests may be reduced to similar
levels regardless of fire intensity. Conversely, post-fire NPP in forests dominated by fire resistant and mixed species decreased with increasing FRP or FRE. In some cases, this dose-response relationship persisted for more than a decade post-fire, highlighting a legacy effect of fire intensity on post-fire C dynamics in these forests.

## 1 Introduction

Forested ecosystems cover ~30% of Earth's land surface and serve as one of the largest terrestrial carbon (C) sinks (Bonan
2008, IPCC 2013). Dynamic ecological processes such as wildfires impact this sink through direct C emissions from combustion, loss of C uptake through tree mortality, decomposition processes, and sequestration of black C within forest soils (Bowman et al., 2009; Brewer et al., 2013; Tinkham et al., 2016). Recent research has demonstrated that greater fire intensity impairs the ability of surviving saplings to photosynthesize (Smith et al., 2016, 2017). However, at landscape spatial scales, while many studies have examined and projected post-fire trends in forest productivity (Goetz et al., 2007; Hicke et al., 2003;
Kashian et al., 2006; Romme et al., 2011), none have evaluated relationships between the fire intensity and those trends.



Characterization of such relationships is essential given both lower fuel moisture (Gergel et al., 2017) and higher fire activity (intensity, frequency, and size) are predicted in North American forested ecosystems under anthropogenic climate change (Balshi et al., 2009, IPCC 2013, Barbero et al., 2015, Abatzoglou and Williams, 2016, Bowman et al., 2017).

Recent studies have observed that increasing fire radiative energy (FRE: J) and peak fire radiative power (FRP: W) incident
on trees results in reduced tree growth and increased mortality (Smith et al., 2017; Sparks et al., 2016, 2017). FRP is the instantaneous radiative flux, which is strongly related to common field-based fire intensity metrics (Kaufman et al., 1996; Kremens et al., 2012; Sparks et al., 2017), and its temporal integral is FRE. These are two of the most commonly used metrics to quantify fire intensity from satellite remote sensing products (Andela et al., 2015; Freeborn et al., 2016; Heward et al., 2013; Roberts et al., 2011; Smith and Wooster, 2005). Under controlled experiments on saplings, a toxicological "dose-response"
relationship was observed, whereby increasing FRE resulted in decreasing net photosynthesis in surviving *Pinus contorta* and *Larix occidentalis* saplings (Smith et al., 2016, 2017) and increased mortality 1-year post-fire (Sparks et al., 2016). Furthermore, Sparks et al. (2017) observed decreasing radial growth in mature Pinus ponderosa 1.5 years post-fire with increasing peak FRP. These findings suggest that there is a strong link between measures of fire intensity and subsequent vegetation productivity and mortality.

Prior studies have been limited to the spatial scale of the individual plant and only up to ~1.5 years following fire treatments. They have also not evaluated how relative fire resistance, or the ability of a tree species to withstand and survive heat-induced damage from fire (Midgley et al., 2011; Starker, 1934; VanderWeide and Hartnett, 2011), may affect the observed dose-response relationship. Numerous studies have linked morphological traits to post-fire survival; thicker bark, deep rooting depth, and a high, open tree crown have all been identified as characteristics that increase relative fire resistance of a tree (Fischer
and Bradley, 1987; Harrington, 2013; He et al., 2012; Keeley, 2012; Midgley et al., 2011; Ryan and Reinhardt, 1988; Starker, 1934; VanderWeide and Hartnett, 2011). However, many studies assume a binary response regarding fire impacts on vegetation: either mortality (immediate or delayed) or no physiological effect (Smith et al., 2017). Consequently, there is a need to investigate if dose-response relationships can be quantified at larger spatial and temporal scales and across forest stands dominated with species of varying levels of fire resistance.

Active and post-fire observations from MODIS provide an avenue to expand previous dose-response studies to a landscape spatial scale and across decadal temporal scales. Terra and Aqua satellites can observe active fires up to four times daily at 1 km resolution at nadir (Justice et al., 2002), enabling the coarse integration of FRP over the duration of a fire, described as fire radiative energy (Boschetti and Roy, 2009; Kumar et al., 2011). However, the relatively low temporal resolution results in significant underestimations of FRE when compared with higher temporal resolution sensors (Vermote et al., 2009). MODIS
observations have also enabled global estimations of Gross Primary Production (GPP), the total amount of C fixed by vegetation, and Net Primary Production (NPP), GPP minus C losses to respiration, when used in tandem with local meteorological data (Running et al., 2004; Zhao and Running, 2010). These estimates have been critical to understanding C



fluxes and forest disturbances over large spatial extents (Bright et al., 2013; Zhao and Running, 2010). Given the lack of landscape scale studies that quantify fire intensity and species composition impacts on post-fire C dynamics, the objective here was to understand how fire intensity affects post-fire productivity in conifer-dominated forested ecosystems. Our results provide further insight into post-fire C dynamics and a framework for spatiotemporal assessments of fire effects.

In this study, we sought to answer the following questions:

1. What are the relationships between fire intensity (i.e., FRP and FRE) and post-fire forest NPP at spatial scales of large wildland fires?

2. How do these relationships vary over time?

3. How do these relationships vary with species composition?

## 10   2 Methodology

### 2.1 Wildland fire selection

Fifteen wildland fires in the Northern Rocky Mountains, U.S., were selected for this study (Figure 1). Fires were chosen to represent coniferous forest stands ranging from those dominated by fire-resistant species to those dominated by fire-susceptible species. Forests dominated by resistant species typically were composed of *Pseudotsuga menziesii*, *Pinus ponderosa*, *Larix*

*occidentalis*, and lesser quantities of *Abies grandis*. Forests dominated by fire-susceptible species typically were composed of *Picea engelmannii*, *Abies lasiocarpa*, *Pinus contorta*, and lesser quantities of *Pinus albicaulis*. To assess the pre-fire dominant forest cover for each fire, we used the LANDFIRE Existing Vegetation Type (EVT) 30 m product (LANDFIRE 2013). Fire selection was based on the following criteria:

(i) Located in northwestern United States temperate forests to minimize latitudinal climatological gradients;

(ii) Located completely within a designated wilderness or other protected area to minimize confounding factors such as land management disturbance;

(iii) Occur in a closed canopy (mean canopy cover > 60%), conifer-dominated forest to minimize mixed pixels;

(iv) Located in forests where the majority of fire-affected area has not been observed to have burned in the last ~30 years; and

(v) Each fire must have at least 3 years of pre- and post-fire MODIS NPP estimates.


Canopy cover for each fire was determined by aggregating the 30 m National Land Cover Database (NLCD) Percent Tree Canopy product (Homer et al., 2007) to the 1 km spatial resolution of the MODIS products. We used the Landsat-derived





Monitoring Trends in Burn Severity (MTBS) fire polygons to estimate whether a forest had burned since 1984 (Eidenshink et al., 2007). Summary information for each fire is given in Table 1.

## 2.2 MODIS datasets

For each fire, we assessed post-fire NPP trajectories as a function of co-located FRP using MODIS NPP and FRP products.
We used the MOD17A3 version 055 1 km NPP product (kg C m$^{-2}$ yr$^{-1}$) to characterize changes in productivity within and between our study fires. The NPP product is detailed in Running et al., (2004) and Hasenauer et al., (2012). MODIS land cover, FPAR (fraction of photosynthetically active radiation), and LAI (leaf area index) products are used in tandem with meteorological data (incoming PAR, stress scalars for high vapor pressure deficit and temperature) and physiological parameters for different vegetation types to calculate daily GPP. NPP is calculated as the sum of GPP over a year minus
maintenance and growth respiration. We acquired the NPP product from the Numerical Terradynamic Simulation Group (NTSG) at the University of Montana. NPP data for years 2000-2015 were downloaded from the NTSG FTP site and analyzed in the native Sinusoidal equal area projection.

To calculate FRP metrics co-located with each NPP pixel we used the Collection 6 MODIS 1 km Level 2 active fire product that identifies and quantifies active fire detections from NASA Terra (MOD14) and Aqua (MYD14) satellites (Giglio et al.,
2016). MODIS FRP is derived from the linear relationship between mid-infrared (4 µm) spectral radiance and FRP (Wooster et al., 2003) and is affected by several factors, including fire background characterization and atmospheric water vapor (Wooster et al., 2005). The 1 km spatial resolution MOD14 and MYD14 products provide an active fire mask showing which pixels contain active fire as well as the date, time, FRP, and other ancillary data (Giglio 2010). Following Boschetti and Roy, (2009), MOD14 and MYD14 data were projected to the 1 km MODIS sinusoidal projection using nearest neighbor resampling.
Importantly, this methodology accounts for increasing MODIS pixel size at large scan angles (Wolfe et al., 1998) so that the location of fire detections and total FRP is preserved post-reprojection. The resulting range of FRP for all fires was comparable to FRP observed in other closed-canopy temperate forests (Giglio et al., 2006; Heward et al., 2013). We used MTBS fire perimeters and metadata to screen any fire detections that were not co-located with recorded fire events spatially and temporally. Fire detections outside the MTBS perimeter were included in the subsequent analysis if they were closer than 1000
m from the MTBS fire boundary (as fires can occur anywhere in the 1 km FRP product).

## 2.3 Data analysis

We calculated FRP distributional statistics (peak, 90$^{th}$ percentile, mean) and FRE for each fire-affected pixel. FRE was calculated following Boschetti and Roy, (2009), where FRP values are integrated over time assuming that FRP varies linearly between observations. FRP and FRE metrics were chosen as they have been demonstrated to have a dose-response relationship
with conifer growth and mortality (Smith et al., 2016, 2017; Sparks et al., 2017). Fire-affected pixels were grouped by FRP and FRE percentile classes (0-25, 25-50, 50-75, 75-100) for each fire. Unburned pixels ($n$unburned = $n$FRP percentile group)





were manually selected outside the MTBS fire perimeters to serve as 'control' pixels. Control pixels were selected if they: 1) were within a 5 km buffer of the MTBS perimeter, 2) were in the same forest type as the fire-affected pixels, and 3) had pre-fire mean NPP within ±50 g C m$^{-2}$ y$^{-1}$ of pre-fire mean NPP of fire-affected pixels. Pre- and post-fire NPP were used to calculate the percent deviation from mean pre-fire NPP, or relative NPP, for each pixel ($i,j$) and year ($t$), which was calculated following Eq. (1):

$$Relative\ NPP_t\ (\%) = \frac{(NPP_{i,j,t} - \overline{NPP_{prefire\,(i,j)}})}{NPP_{prefire\,(i,j)}} \tag{1}$$

To account for interannual variability in NPP not caused by the fires we subtracted the unburned (control) pixel values from the burned pixel values (Bright et al., 2013; Goetz et al., 2006). After confirming normality and homogeneity of variances, differences between FRP percentile classes were assessed using ANOVA with a post hoc Tukey's honest significant difference test ($\alpha = 0.05$). Recovery time for the fire affected pixels was also assessed and was defined as the time necessary for post-fire total NPP to equal or surpass mean pre-fire NPP at the same location.

## 3 Results

### 3.1 Fire intensity differences between forest types

Fires in forests dominated by fire-susceptible species were more intense, in terms of FRP metrics and FRE, than those dominated by a mix of species or fire-resistant species (Figure 2). Average peak FRP values were 212.5 ± 292.3 MW km$^{-2}$ in forests dominated by fire-susceptible species, 150.8 ± 117.4 MW km$^{-2}$ in mixed forests and 100.2 ± 77.5 MW km$^{-2}$ in forests dominated by fire-resistant species (Figure 2a). Similarly, average 90$^{th}$ percentile FRP was 159.7 ± 90.8 MW km$^{-2}$ in forests dominated by fire-susceptible species, 109.4 ± 78.6 MW km$^{-2}$ in mixed forests and 75.7 ± 53.6 MW km$^{-2}$ in forests dominated by fire-resistant species (Figure 2a). Mean FRP values ranged from 83.5 ± 43.8 MW km$^{-2}$ in forests dominated by fire-susceptible species, 58.9 ± 45.9 MW km$^{-2}$ in mixed forests and 43.5 ± 29.7 MW km$^{-2}$ in forests dominated by fire-resistant species (Figure 2a). Average FRE followed the same trend as FRP (Figure 2b). FRE was greater in forests dominated by fire-susceptible species (4,158.3 ± 2,630.4 MJ km$^{-2}$) than those dominated by a mix (2,685.2 ± 1,660.5 MJ km$^{-2}$) or fire-resistant species (2,095.2 ± 1,389.5 MJ km$^{-2}$).

### 3.2 Higher fire intensity results in lower post-fire NPP

For fires that occurred in forests dominated by a mix of species or fire-resistant species, higher FRP or FRE magnitude led to lower post-fire NPP (Figure 3, columns 1-2). This dose-response relationship was strongest (more significant differences between the relative NPP across FRP and FRE percentile classes) one year post-fire, where mean relative NPP decreased with increasing FRE (Figure 3, row 1) and FRP (Figure 3, rows 2-4). In forests dominated by fire-resistant species, there was a stronger dose-response pattern for relative NPP grouped by FRE percentile class rather than FRP percentile class (Figure 3, column 1). For mixed forests, the dose-response pattern was the same regardless of whether relative NPP was grouped by FRE



or FRP percentile classes. The observed dose-response relationship for these forest types persisted for up to >8 years post-fire, especially in forests dominated by mixed species (Figure 3, column 2). The dose-response relationship was much weaker in forests dominated by fire-susceptible species (Figure 3, column 3). There were few differences between percentile classes with only the highest FRE percentile class displaying lower relative NPP compared with the other percentile classes.

Maximum relative NPP loss occurred at one year post-fire for all fires and differed by species composition. Generally, mixed stands consisting of fire-susceptible and fire-resistant species had the largest relative post-fire NPP losses with an average loss of 40.7% (-216.7 g C m$^{-2}$ yr$^{-1}$), followed by stands that were dominated by fire-susceptible species with an average loss of 33.9% (-154.8 g C m$^{-2}$ yr$^{-1}$). Stands dominated by fire resistant species had the smallest average loss of 23.3% (-126.8 g C m$^{-2}$ yr$^{-1}$).

### 3.3 Recovery and trajectories of post-fire NPP

Post-fire observations ranged from 4-12 years post-fire (average 8.4 years) for the fifteen fires, however, only the lowest FRP class of one fire (2006 South Fork Fire) had recovered to pre-fire NPP levels at the end of the observational period (~9 years post-fire). Generally, recovery trajectories were linear for all fire resistance groups, except for a few fires where NPP began decreasing again around 2011 (Supplemental Figures S1-S4).

### 4 Discussion

### 4.1 Higher fire intensity results in lower post-fire NPP

To date, research has largely analyzed post-fire forest productivity with fire as a binary predictor variable (presence-absence). In the current study, we applied a dose-response methodology that has been demonstrated at the tree-scale (Smith et al., 2016, 2017, Sparks et al., 2016, 2017) to large fires using landscape remote sensing datasets. A dose-response relationship between FRP or FRE and NPP was shown in forests dominated by fire-resistant species and mixed species (Figures 3, columns 1-2). Forests that were dominated by fire-susceptible species were not as sensitive to FRP or FRE, indicating that NPP in these forests may be reduced to similar levels regardless of fire intensity (Figure 3, column 3). Additionally, forests dominated by fire resistant species had lower post-fire relative NPP losses compared to those dominated by fire-susceptible species or a mix (Figure 3). These data are congruent with evidence at the tree scale where trees that do not develop fire resistant traits, such as thick bark, have a higher probability of fire-induced damage and mortality (Midgley et al., 2011; Ryan and Reinhardt, 1988; VanderWeide and Hartnett, 2011). NPP loss at two years post-fire (~19-152 g C m$^{-2}$ yr$^{-1}$) in forests dominated by fire-resistant species is comparable to two-year post-fire aboveground NPP differences between unburned and burned temperate *Pinus ponderosa* forest stands (~83-148 g C m$^{-2}$ yr$^{-1}$), estimated using field measurements (Irvine et al., 2007).

30



There was considerable variability in the dose-response relationships within each fire resistance grouping, which could potentially be attributed to differences in stand structure and age as well as differing proportions of burned and unburned area within each NPP pixel (mixed pixels). Previous studies have indicated that smaller trees are more susceptible to fire-induced mortality than larger trees (Hood et al., 2007). Additionally, there is evidence that similar FRP doses can lead to widely

different growth responses depending on tree age (Smith et al., 2017; Sparks et al., 2017). For example, 2.5-year-old *Pinus contorta* and *Larix occidentalis* saplings exposed to highly controlled laboratory surface fires (peak FRP ranged from 4.1-12.9 kW m$^{-2}$) had radial growth at 1-year post-fire that was -2.5% to -20% of unburned saplings (Smith et al., 2017). In contrast, a similar range of peak FRP (0.2-16.3 kW m$^{-2}$) was observed in prescribed fires in 34-year-old *Pinus ponderosa* stands, but resulted in tree radial growth that was -10% to -45% of unburned tree radial growth at 1.5 years post-fire (Sparks et al., 2017).

The forests analyzed in this study likely had highly heterogeneous stand structures and ages within each 1 km MODIS pixel, which could lead to highly heterogenous fire behaviour and vegetation response within a pixel. While previous studies mainly assessed surface fire impacts on trees, it is likely that areas within each of the fires in this study had complete overstory removal via crown fire. Variability in fire behaviour can also lead to unburned islands within each fire perimeter. Previous studies have quantified unburned proportions within MTBS perimeters ranging from ~10-25% of within perimeter area (Kolden et al., 2015;

Meddens et al., 2016), which could lead to more mixed pixels (pixels containing burned and unburned forest). These sub-pixel differences could lead to widely different patterns of mortality and recovery and mask any pixel-scale dose-response relationship.

The observed dose-response relationship was also affected by the number of MODIS FRP observations per pixel. The mean

number of FRP observations per pixel decreased from fires with a clear dose-response relationship (4.8 ± 0.6 obs. pixel$^{-1}$) to those with a weak relationship (2.9 ± 0.1 obs. pixel$^{-1}$). This pattern could be attributed to the long temporal intervals between consecutive satellite overpasses and consequently, a poorer overall characterization of the fire behavior for a particular pixel (Giglio, 2007). This factor could also account for the slight differences observed between FRP metrics and FRE, as the long intervals between consecutive satellite overpasses have a high probability of missing increased fire activity associated with

peak and 90[th] percentile FRP (Giglio, 2007).

### 4.2 Recovery and trajectories of post-fire NPP

Despite an average post-fire observational period of 8.4 years across all fires, only the lowest FRP or FRE percentile class of one fire (2006 South Fork Fire) had recovered to pre-fire NPP levels ~9 years post-fire. Other studies that have used remote sensing observations reported recovery time ranging from 5 years (Goetz et al., 2006) to 9 years (Hicke et al., 2003) in boreal

forests. Likewise, chronosequence studies in boreal forest have estimated recovery to be ~10 years (Amiro et al., 2010). The results from this study are consistent with observations showing large differences in productivity between burned and unburned forest stands at time periods greater than 10 years post-fire (Dore et al., 2008). The convergence of some of the NPP trajectories could be attributed to rapid recovery and colonization of fire affected areas by understory species (Goetz et al., 2006). The




forests in the current study occur in areas where rapid post-fire colonization by shrub and herbaceous species is common (Jorgensen and Jenkins, 2011), which could make NPP appear to recover more rapidly in areas where the forest overstory has been removed (Bright et al., 2013).

### 4.3 Conceptual framework for assessing spatiotemporal post-fire effects

The results presented in this work, building upon tree-scale studies, provide a framework that links fire intensity to post-fire changes in individual tree and forest growth/productivity (Figure 4). In this conceptual system, several post-fire recovery pathways exist for trees/forests depending on the initial fire intensity. We hypothesize that higher intensity fires cause trees to incur more damage, which can lead to rapid mortality if trees have insufficient resources to repair physiological function in the weeks and months following a fire. The highest fire intensities lead to the greatest losses in physiological function and net primary productivity in surviving trees (Smith et al., 2017; Sparks et al., 2017) as well as the highest probability of delayed mortality in the years after a fire (Sparks et al., 2016). Moderate levels of fire intensity cause enough damage to decrease growth/productivity and alter a tree's vulnerability to secondary mortality agents (e.g. insects, disease and drought). Vulnerability may be lessened if permanent defensive structures, such as resin ducts in *Pinus* species used for expelling bark beetles, are induced by the fire (Hood et al., 2015; Sparks et al., 2017). On the contrary, fire may make trees more susceptible to secondary mortality agents if the photosynthetic machinery of trees is sufficiently impaired (Davis et al., 2012). Trees experiencing low intensity fires will likely have reduced growth, but a higher probability of surviving than trees subjected to higher fire intensities. For any post-fire pathway, trees in better physiological condition or those exposed to fewer environmental stressors will likely experience a lower impact to post-fire growth and a lower probability of mortality.

### 4.4 Limitations

The dose-response relationship we observed between FRP or FRE and post-fire NPP does not necessarily mean this methodology can now be directly applied to the characterization of landscape-scale C dynamics; several limitations are obvious. First, this study analyzed fires that occurred in forests with little-to-no management disturbance. Applying this methodology to managed forests may produce significantly different results as land management disturbances (e.g. timber harvest, urban development) may alter the dose-response relationship between FRP and NPP. Second, in forests with canopy cover less than 100%, or where fire has completely removed the overstory, MODIS observes reflectance from overstory and understory forest vegetation. Understory vegetation that recovers rapidly could alter the magnitude of post-fire NPP reduction and make it appear that the overstory recovers more (or less) quickly. Finally, due to the fact that MODIS FRP observations per pixel were generally low for fires in this region (mean = 3.9 observations pixel$^{-1}$), caution should be used when interpreting results and comparing to other ground and remote sensing based measurements.



## 5 Conclusions

Through the use of remotely sensed fire radiative power and net primary productivity, we demonstrate that increasing doses of FRP and FRE lead to decreasing post-fire net primary productivity in coniferous forests, especially those dominated by fire-resistant tree species. This dose-response relationship appears to have a legacy effect on C dynamics, in some cases lasting for

greater than a decade post-fire. Species composition also influenced the magnitude of post-fire NPP loss, highlighting the importance of the relative fire resistance of forest species in accounting for post-fire C dynamics. While this dose-response relationship is promising, our results indicate that a low number of FRP observations diminish the detectability of this relationship. Despite post-fire observations ranging up to 12 years, most of the forests had not recovered to pre-fire productivity levels, which agrees with field observations showing large differences in productivity between burned and unburned temperate

forests up to a decade post-fire (Dore et al., 2008). Ultimately, this study extends prior tree-scale dose-response studies and presents a framework for using fire radiative metrics to quantifying long-term post-fire effects, such as reduction and recovery of NPP, at the landscape spatial scale.

## Acknowledgements

Partial funding for Sparks was provided by the Joint Fire Science Program under GRIN Award 16-2-01-09. This material is based upon work supported by the National Science Foundation under grant no. DMS-1520873, and the Department of the Interior Northwest Climate Science Center (NW CSC) through a Cooperative Agreement [G14AP00177] from the United States Geological Survey (USGS). Its contents are solely the responsibility of the authors and do not necessarily represent the views of the NW CSC or the USGS. This manuscript is submitted for publication with the understanding that the United States

Government is authorized to reproduce and distribute reprints for Governmental purposes.



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



**Table 1:** Summary of the fifteen fires analyzed in this study.

| Fire Name | Size (ha) | Location | Dominant Conifer Species | Ignition Date |
|---|---|---|---|---|
| Ahorn | 18,778 | Montana | *Pseudotsuga menziesii, Pinus ponderosa, Larix occidentalis, Picea engelmannii, Abies lasiocarpa, Pinus albicaulis* | June 28, 2007 |
| Arnica | 4,556 | Wyoming | *Pinus contorta* | September 23, 2009 |
| Bridge | 15,116 | Idaho | *Picea engelmannii, Abies lasiocarpa* | July 18, 2007 |
| Columbine | 7,115 | Wyoming | *Picea engelmannii, Abies lasiocarpa, Pinus albicaulis* | August 9, 2007 |
| East | 7,145 | Wyoming | *Picea engelmannii, Abies lasiocarpa, Pinus albicaulis, Pinus contorta* | August 8, 2003 |
| Fawn Peak | 31,870 | Washington | *Pseudotsuga menziesii, Pinus ponderosa, Larix occidentalis, Picea engelmannii, Abies lasiocarpa, Pinus albicaulis* | June 30, 2003 |
| Fool Creek | 22,186 | Montana | *Picea engelmannii, Abies lasiocarpa, Pinus albicaulis* | June 28, 2007 |
| Little Salmon | 13,598 | Montana | *Pseudotsuga menziesii, Pinus ponderosa, Larix occidentalis, Picea engelmannii, Abies lasiocarpa* | July 18, 2003 |
| Meriwether | 7,762 | Montana | *Pseudotsuga menziesii, Pinus ponderosa, Larix occidentalis* | July 21, 2007 |
| North Fork | 6,774 | Oregon | *Pseudotsuga menziesii, Pinus ponderosa* | August 1, 2009 |
| Saddle | 12,706 | Idaho | *Pseudotsuga menziesii, Pinus ponderosa, Larix occidentalis, Picea engelmannii, Abies lasiocarpa* | August 18, 2011 |
| Sawmill | 6,015 | Montana | *Pseudotsuga menziesii, Pinus ponderosa, Larix occidentalis* | July 13, 2007 |
| Shower Bath | 19,911 | Idaho | *Pseudotsuga menziesii, Pinus ponderosa* | July 17, 2007 |
| South Fork | 11,494 | Idaho | *Pseudotsuga menziesii, Pinus ponderosa* | August 7, 2006 |
| Tatoosh | 20,185 | Washington | *Pseudotsuga menziesii, Pinus ponderosa, Larix occidentalis, Picea engelmannii, Abies lasiocarpa, Pinus albicaulis* | August 22, 2006 |





**Figure 1:** Location of study fires overlaid on current distribution of U.S. forest types classified using relative fire resistance information in the literature and the LANDFIRE Existing Vegetation Type (EVT) 30-m product.



**Figure 2:** Fire radiative power (a) and fire radiative energy (b) distributional statistics grouped by dominant forest composition (fire-resistant to fire-susceptible). Black arrows indicate mean values.

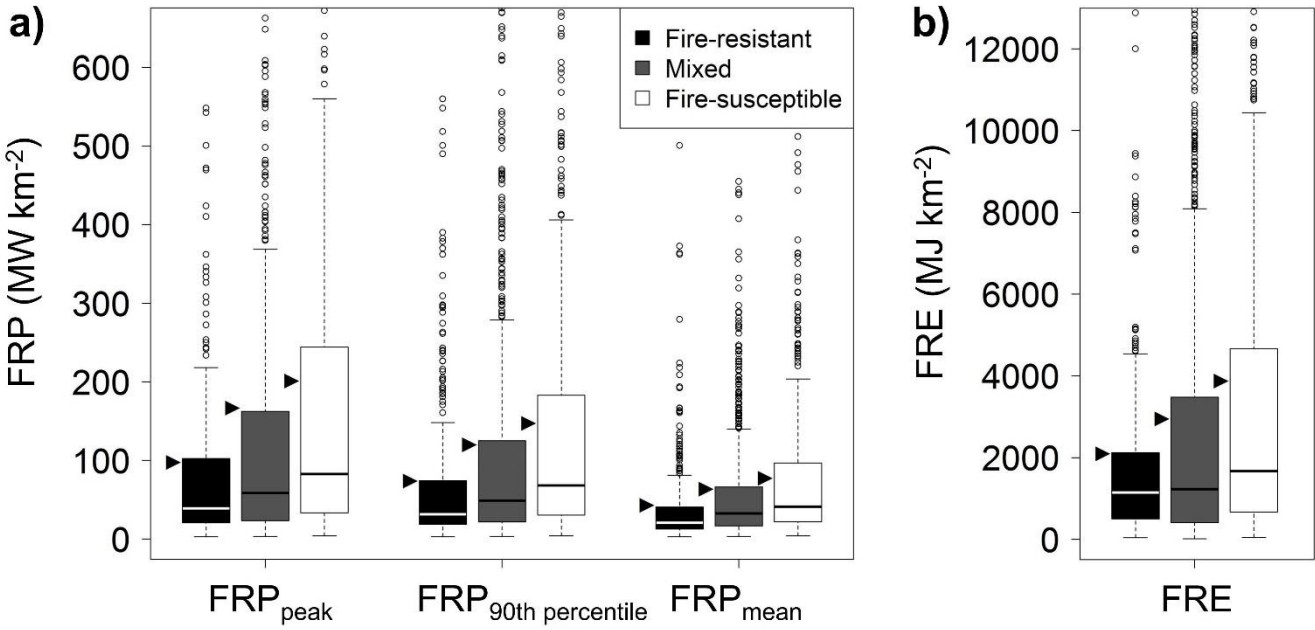





**Figure 3.** FRP and FRE dose impacts on NPP response observed in forests dominated by species varying from fire-resistant to fire-susceptible (first column – third column). Mean relative NPP is grouped by FRP and FRE percentile classes and shading represents 95% confidence intervals.





**Figure 4.** Conceptual framework for quantifying impacts of fire intensity on physiology, growth, and vulnerability of coniferous forests.

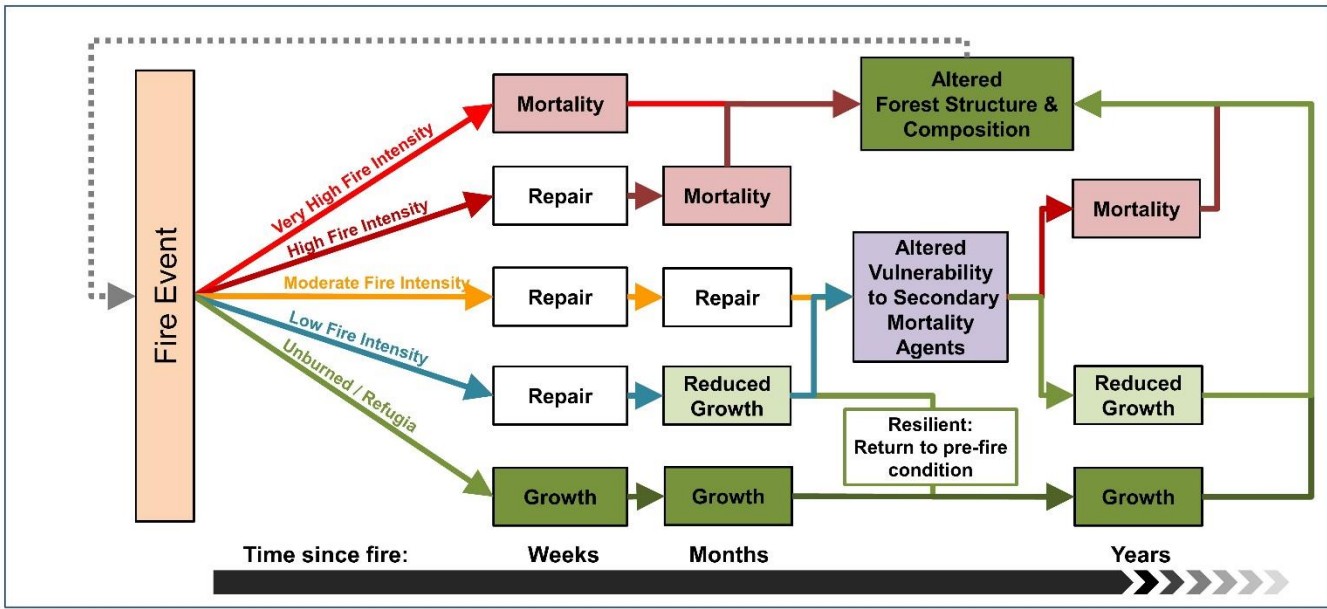