# Peer review of "Fire intensity impacts on post-fire temperate coniferous forest net primary productivity"

_Biogeosciences, 2017_

## Referee Comment (RC1) · Anonymous Referee #1 · 29 Sep 2017

The manuscript by Sparks and colleagues examined how forest composition and fire intensity affected forest net primary productivity (NPP) following fire. The authors argue that higher fire intensity leads to progressively larger reduction in post-fire NPP among fire resistant and mixed-resistance communities, while fire intensity had little effect on the magnitude of NPP change in fire susceptible communities. The manuscript is well written and presents an analysis that provides novel insight into forest carbon dynamics following fire in a region where fire activity is likely to intensify over the coming century due to regional warming and drying. As detailed below, it seems there are several aspects of this analysis could be refined to further improve its rigor.

**Primary comments**

1. The manuscript states that, "Fire-affected pixels were grouped by FRP and FRE percentile classes (0-25, 25-50,…) for each fire" and then changes in post-fire NPP were evaluated among these percentile classes across all fires within a forest type. Why group pixels by fire-specific percentile class rather than by the absolute magnitude of fire intensity? Perhaps I am misunderstanding the approach, but let's say there are two fires of contrasting intensity, both of which occur in a fire-susceptible forest type. In this forest type, about 50% of pixels had FRE < 2000 MJ km$^{-2}$ and about 50% of pixels had FRE between 2000 and 12000 MJ km$^{-2}$ (figure 2). If the low-intensity fire only experienced FRE < 2000 MJ km$^{-2}$ and the high intensity fire only experienced FRE > 2000 MJ km$^{-2}$, then what happens when the pixels within each fire are grouped by the fire-specific percentile class and then these classes are subsequently grouped across fires? The 75-100$^{th}$ percentile class for the low severity fire might have FRE of, say, 1000-2000 MJ km$^{-2}$, whereas this same percentile class for the high severity fire might encompass areas where the FRE was > 10,000 MJ km$^{-2}$. You might expect a very different post-fire trajectory of NPP between these two fires for the same percentile class, but at present these would get grouped together, correct? This might somewhat explain why you don't see any difference between percentile classes in post-fire NPP trajectory for the fire-susceptible forest type.
2. The description of the statistical analysis is vague and the results do not present any statistics. Also, how do account for taking multiple pixels from the same fire and using them as independent samples, when in fact they are not independent?

**Secondary comments**

1. The researchers frequently note that there are dose-response relationships between fire intensity and post-fire changes in NPP. Given this focus, it would be worth including a figure that more explicitly shows this relationship. The figure could show the change in NPP one year after fire as a function of fire intensity for each of the three forest types.
2. Could it be that fire intensity is higher in fire-susceptible forests than mixed or fire-resistant forests not solely because of differences in trait characteristics, but rather because there is more biomass (fuel) in these forests? It could be worth normalizing fire intensity by forest biomass to see whether fire intensity per unit of fuel differs between these three broad forest types. The National Biomass and Carbon Data set 2000 (NBCD2000) could be a useful source of information for this endeavor (https://daac.ornl.gov/cgi-bin/dsviewer.pl?ds_id=1161)

3. The manuscript includes figures showing the relative change in NPP following fire, but not the absolute change in NPP. It would be informative to show how the absolute magnitude of NPP changes after fire.
4. Does including the $FRP_{90th}$ percentile add to the story? It seems somewhat redundant given the inclusion of $FRP_{peak}$ and $FRP_{mean}$.

**Line specific comments**
1. Page 4, line 31: What does "Unburned pixels ($n$unburned = $n$FRP percentile group)…" mean? Does this mean that you selected the same number of unburned pixels as there were pixels in the percentile group?
2. Page 5, lines 14-25: The researchers present the average and variation (presumably SD, but not defined) in fire intensity metrics for each forest type; however, Figure 2 shows that these metrics are very non-normally distributed. Consequently, mean and standard deviation are not appropriate summary statistics. The median and interquartile range would be more appropriate.
3. Page 5, lines 29-30: the researchers state that, "in forests dominated by fire-resistant species, there was a stronger dose-response pattern for relative NPP grouped by FRE percentile class rather than FRP percentile class." This pattern is not particularly evident looking at figure 3. I would suggest providing additional evidence, or removing the statement.
4. Page 6, line 4: "The dose-response relationship was much weaker in forests dominated by fire susceptible species. There were few differences between percentile classes with only the highest FRE percentile class displaying lower relative NPP compared with other percentile classes." Is this based on a qualitative comparison, or statistical analysis?
5. Page 6, line 14: The authors state that "generally, recovery trajectories [in NPP] were linear for all fire-resistant groups, except for a few fires were NPP begin decrease again around 2011." Looking at the supplemental figures, it appears that many, if not most, of the fires show non-linear changes in NPP after fire.
6. Page 7, lines 19-21: The authors note that the number of MODIS FRP observation differed between "fires with a clear this up – response relationship" and those with a "weak relationship." Does this suggest that there were differences in the number of MODIS FRP observations between forest types? Perhaps clarify what is meant by a clear relationship versus a weak relationship.
7. Page 8, Conceptual framework: The following citations could bolster this section:
    i. Michaletz, S. T., E. Johnson, and M. Tyree. 2012. Moving beyond the cambium necrosis
        1. hypothesis of post-fire tree mortality: cavitation and deformation of xylem in forest fires. New Phytologist **194**:254-263.
    ii. van Mantgem, P. J., J. C. Nesmith, M. Keifer, E. E. Knapp, A. Flint, and L. Flint. 2013.
        1. Climatic stress increases forest fire severity across the western United States. Ecology Letters **16**:1151-1156.
8. Page 8, line 22: Always hesitant to say things are "obvious" in a paper.
9. Supplemental figures: The Saddle fire appears to be missing the vertical line denoting the year in which the fire occurred. Also, what do the plotting characters and error represent in these figures? Mean and standard deviation?

---

## Referee Comment (RC3) · Anonymous Referee #2 · 30 Oct 2017

This is an interesting paper detailing how NPP varies with fire severity across 15 large fires in the western U.S. MODIS satellite data at the 1-km pixel scale was used, giving a coarse view of fire severity effects on productivity. The paper addresses relevant scientific questions, presents novel results, and reaches substantial conclusions. However, some aspects of the paper, both major and minor, could be improved. General and specific comments follow.

General comments:

–Freeborn et al. 2014 reported that differences in per-pixel FRP measured near-simultaneously have a standard deviation of 27%, and that clumping pixels helps a lot (50-pixel aggregation reduces uncertainty to 5%; citation at end of comments). This seems like a relevant issue for the current study, since it uses pixel-level data. Would

including the uncertainty in the analysis change the results or the interpretation of the results?

–I don't find the conceptual framework (page 8 and Figure 4) to be very strong. The authors state that they are linking individual tree-level processes to fire intensity and forest growth and productivity. But they go on to say in the Limitations section that understory vegetation may recover rapidly and make it appear that the overstory recovers rapidly. It doesn't seem that the authors can actually say much about individual tree mortality, given the heterogeneity of fires on the ground, the large size of the pixels being used, and the lack of on-the-ground severity measurements. Couldn't it be that shrubs are what are responding post-fire rather than trees?

–Finally, I agree with the first reviewer in questioning why the authors grouped the FRP and FRE into percentile classes, because then it's difficult to compare actual FRP and FRE in terms of their effect on NPP across fires- you've limited the analysis to within-fire differences. Similarly, I also question why relative NPP rather than absolute NPP is shown in the supplemental figures. Are there are interesting absolute differences among forest types?

Specific comments:

–Page 4, Line 1: MTBS only includes fires 1000 acres and bigger: are the authors able to verify through other data sources that these areas haven't burned since 1984? Does it matter?

–Page 4, MODIS datasets: Was FRP available for all pixels inside the MTBS perimeters?

–Page 5, section 3.1: All of the numbers in this paragraph could go into a table and it might be easier to read.

–Page 5, Line 7: It's mentioned here that other things besides fire may contribute to NPP variability, but I don't think it was mentioned again. It's worth noting in the

discussion whether climate or other factors might play a role in post-fire recovery of NPP.

Technical Corrections: –Page 3, Line 12: Some of the sites are not in the Northern Rocky Mountains. –Page 3, Lines 19-24: Pick past or present tense to be consistent throughout. –Page 3, Line 26: "Canopy cover for each fire"- do you mean pre-fire canopy cover?

Citation: Freeborn, P.H. M.J. Wooster, D.P. Roy, and M.A. Cochrane. 2014. Quantification of MODIS fire radiative power (FRP) measurement uncertainty for use in satellite-based active fire characterization and biomass burning estimation. Geophysical Research Letters 41(6):1988-1994.

---

## Author Comment (AC1) · 29 Nov 2017

Overall response: We would like to thank referee #1 for the supportive comments on the manuscript. We will implement their suggestions in our revised manuscript. Below we respond to each of the comments individually.

Anonymous Referee #1 Received and published: 29 September 2017

The manuscript by Sparks and colleagues examined how forest composition and fire intensity affected forest net primary productivity (NPP) following fire. The authors argue that higher fire intensity leads to progressively larger reduction in post-fire NPP among fire resistant and mixed-resistance communities, while fire intensity had little effect on the magnitude of NPP change in fire susceptible communities. The manuscript is well

written and presents an analysis that provides novel insight into forest carbon dynamics following fire in a region where fire activity is likely to intensify over the coming century due to regional warming and drying. As detailed below, it seems there are several aspects of this analysis could be refined to further improve its rigor.

Primary comments 1. The manuscript states that, "Fire-affected pixels were grouped by FRP and FRE percentile classes (0-25, 25-50,: : :) for each fire" and then changes in post-fire NPP were evaluated among these percentile classes across all fires within a forest type. Why group pixels by fire-specific percentile class rather than by the absolute magnitude of fire intensity? Perhaps I am misunderstanding the approach, but let's say there are two fires of contrasting intensity, both of which occur in a firesusceptible forest type. In this forest type, about 50% of pixels had FRE < 2000 MJ km-2 and about 50% of pixels had FRE between 2000 and 12000 MJ km-2 (figure 2). If the low-intensity fire only experienced FRE < 2000 MJ km-2 and the high intensity fire only experienced FRE > 2000 MJ km-2, then what happens when the pixels within each fire are grouped by the fire-specific percentile class and then these classes are subsequently grouped across fires? The 75-100th percentile class for the low severity fire might have FRE of, say, 1000-2000 MJ km-2, whereas this same percentile class for the high severity fire might encompass areas where the FRE was > 10,000 MJ km-2. You might expect a very different post-fire trajectory of NPP between these two fires for the same percentile class, but at present these would get grouped together, correct? This might somewhat explain why you don't see any difference between percentile classes in post-fire NPP trajectory for the fire-susceptible forest type.

R1: We thank the reviewer for this comment, we realize this section was poorly worded and it will be clarified in the revised version. Percentile classes were based on absolute magnitude of FRE (or FRP) by forest type, not for each individual fire.

2. The description of the statistical analysis is vague and the results do not present any statistics. Also, how do account for taking multiple pixels from the same fire and using them as independent samples, when in fact they are not independent?
R2: We are confused by this comment. The reviewer does not provide any specific example of what "statistics" should be included and specifically what section/paragraph is vague. However, in response to reviewer #2 (comment #6) we will add a table of population parameters, given we are using a census of fire-affected pixels (i.e. all of the pixels within the fire perimeter are considered in the analysis, not a sample).

Secondary comments 1. The researchers frequently note that there are dose-response relationships between fire intensity and post-fire changes in NPP. Given this focus, it would be worth including a figure that more explicitly shows this relationship. The figure could show the change in NPP one year after fire as a function of fire intensity for each of the three forest types.

R3: We think this is an excellent comment and will include a 1-year post-fire doseresponse figure in the revised manuscript.

2. Could it be that fire intensity is higher in fire-susceptible forests than mixed or fireresistant forests not solely because of differences in trait characteristics, but rather because there is more biomass (fuel) in these forests? It could be worth normalizing fire intensity by forest biomass to see whether fire intensity per unit of fuel differs between these three broad forest types. The National Biomass and Carbon Data set 2000 (NBCD2000) could be a useful source of information for this endeavor (https://daac.ornl.gov/cgi-bin/dsviewer.pl?ds\_id=1161)

R4: We thank the reviewer for this suggestion. We downloaded this dataset and found that, on average, estimated biomass did not exactly match trends in fire intensity (FRP, FRE). Average biomass per unit area (Mg ha-1) decreased from mixed (157 Mg ha-1) to fire-susceptible (129 Mg ha-1) to fire resistant (110 Mg ha-1), whereas fire intensity (FRP, FRE) decreased from fire-susceptible to mixed to fire resistant. We will include this data in the table proposed by reviewer #2 (comment #6), and in an expanded discussion section (along with other potential drivers – e.g. climate, forest structural differences between the three types).
3. The manuscript includes figures showing the relative change in NPP following fire, but not the absolute change in NPP. It would be informative to show how the absolute magnitude of NPP changes after fire.

R5: We agree with this suggestion and will modify Figure 3 to show absolute change in NPP in the revised manuscript.

4. Does including the FRP90th percentile add to the story? It seems somewhat redundant given the inclusion of FRPpeak and FRPmean.

R6: We agree with the reviewer and will remove it from Figure 3.

Line specific comments 1. Page 4, line 31: What does "Unburned pixels (nunburned = nFRP percentile group): : :" mean? Does this mean that you selected the same number of unburned pixels as there were pixels in the percentile group?

R7: That is correct – we will clarify this in the revised manuscript text.

2. Page 5, lines 14-25: The researchers present the average and variation (presumably SD, but not defined) in fire intensity metrics for each forest type; however, Figure 2 shows that these metrics are very non-normally distributed. Consequently, mean and standard deviation are not appropriate summary statistics. The median and interquartile range would be more appropriate.

R8: We will correct this in the revised manuscript and display median and IQR.

3. Page 5, lines 29-30: the researchers state that, "in forests dominated by fire-resistant species, there was a stronger dose-response pattern for relative NPP grouped by FRE percentile class rather than FRP percentile class." This pattern is not particularly evident looking at figure 3. I would suggest providing additional evidence, or removing the statement.

R9: We will remove this statement.

4. Page 6, line 4: "The dose-response relationship was much weaker in forests dom-
inated by fire susceptible species. There were few differences between percentile classes with only the highest FRE percentile class displaying lower relative NPP compared with other percentile classes." Is this based on a qualitative comparison, or statistical analysis?

R10: The revised text will clarify that this is a qualitative comparison.

5. Page 6, line 14: The authors state that "generally, recovery trajectories [in NPP] were linear for all fire-resistant groups, except for a few fires were NPP begin decrease again around 2011." Looking at the supplemental figures, it appears that many, if not most, of the fires show non-linear changes in NPP after fire.

R11: This is a valid point. The text will be corrected to say some fires had linear recovery trajectories, while most did not.

6. Page 7, lines 19-21: The authors note that the number of MODIS FRP observation differed between "fires with a clear this up – response relationship" and those with a "weak relationship." Does this suggest that there were differences in the number of MODIS FRP observations between forest types? Perhaps clarify what is meant by a clear relationship versus a weak relationship.

R12: The revised text will clarify this in terms of how dose-response studies are commonly analyzed and reported (e.g. Ruberg 1995). Specifically, the presence or absence of significant differences between fire intensity classes (and the unburned 'control' pixels) will be emphasized.

7. Page 8, Conceptual framework: The following citations could bolster this section: i. Michaletz, S. T., E. Johnson, and M. Tyree. 2012. Moving beyond the cambium necrosis 1. Hypothesis of post-fire tree mortality: cavitation and deformation of xylem in forest fires. New Phytologist 194:254-263. ii. van Mantgem, P. J., J. C. Nesmith, M. Keifer, E. E. Knapp, A. Flint, and L. Flint. 2013. 1. Climatic stress increases forest fire severity across the western United States. Ecology Letters 16:1151-1156. BGD
R13: Thanks for the suggestions, these will be added.

8. Page 8, line 22: Always hesitant to say things are "obvious" in a paper.

R14: This will be re-worded in the revised manuscript.

9. Supplemental figures: The Saddle fire appears to be missing the vertical line denoting the year in which the fire occurred. Also, what do the plotting characters and error represent in these figures? Mean and standard deviation?

R15: This will be corrected and clarified in the revised manuscript.

Response references: Ruberg, S.J., 1995. Dose response studies II. Analysis and interpretation. Journal of biopharmaceutical statistics, 5(1), pp.15-42.

---

## Author Comment (AC2) · 29 Nov 2017

Overall response: We would like to thank referee #2 for the supportive comments on the manuscript. We plan to implement their suggestions in our revised manuscript. Below we respond to each of the comments individually.

Anonymous Referee #2 Received and published: 30 October 2017

This is an interesting paper detailing how NPP varies with fire severity across 15 large fires in the western U.S. MODIS satellite data at the 1-km pixel scale was used, giving a coarse view of fire severity effects on productivity. The paper addresses relevant scientific questions, presents novel results, and reaches substantial conclusions. However, some aspects of the paper, both major and minor, could be improved. General

and specific comments follow.

General comments: 1. Freeborn et al. 2014 reported that differences in per-pixel FRP measured near simultaneously have a standard deviation of 27%, and that clumping pixels helps a lot (50-pixel aggregation reduces uncertainty to 5%; citation at end of comments). This seems like a relevant issue for the current study, since it uses pixel-level data. Would including the uncertainty in the analysis change the results or the interpretation of the results?

R1: We will discuss the potential impacts of this uncertainty in an expanded discussion section of the revised manuscript.

2. I don't find the conceptual framework (page 8 and Figure 4) to be very strong. The authors state that they are linking individual tree-level processes to fire intensity and forest growth and productivity. But they go on to say in the Limitations section that understory vegetation may recover rapidly and make it appear that the overstory recovers rapidly. It doesn't seem that the authors can actually say much about individual tree mortality, given the heterogeneity of fires on the ground, the large size of the pixels being used, and the lack of on-the-ground severity measurements. Couldn't it be that shrubs are what are responding post-fire rather than trees?

R2: Sparks et al. 2016 and Smith et al. 2017 observed mechanistic links between FRP and sapling mortality and productivity. These, and other studies (e.g. Sparks et al. 2017), also collectively demonstrated that the mechanism scaled from the saplings in a laboratory fire to mature trees in stand-scale fires. Prior studies (Ryan and Reinhardt 1988; Hood et al. 2007) had previously reported similar relationships between proxies of fire intensity and mature tree mortality. Although the current paper suggests that this may further scale from the watershed to the regional scale, we agree that this is not yet proven. As such, we will adjust the text to be more circumspect and cautious of a regional scale relationship.

3. Finally, I agree with the first reviewer in questioning why the authors grouped the
FRP and FRE into percentile classes, because then it's difficult to compare actual FRP and FRE in terms of their effect on NPP across fires- you've limited the analysis to within fire differences. Similarly, I also question why relative NPP rather than absolute NPP is shown in the supplemental figures. Are there are interesting absolute differences among forest types?

R3: This section was poorly worded and will be clarified in a revision. Percentile classes were based on absolute magnitude by forest type, not for each individual fire.

Specific comments: 4. Page 4, Line 1: MTBS only includes fires 1000 acres and bigger: are the authors able to verify through other data sources that these areas haven't burned since 1984? Does it matter?

R4: This is a great point. We mapped smaller burned areas within each MTBS polygon using the Normalized Burn Ratio Thermal Index (Holden et al. 2005) computed by Google Climate Engine (climateengine.org) annually from 1984 to the present. Google Climate Engine uses data from Landsat 4, 5, 7, and 8 depending on availability and cloud cover to produce 30 m spatial resolution datasets. Using these data we found that, on average, less than 1.5% of the MTBS polygon area burned between 1984 and the year that each fire burned. We will include this information for each fire in the proposed table (comment #6) in a revised manuscript.

5. Page 4, MODIS datasets: Was FRP available for all pixels inside the MTBS perimeters?

R5: On average, FRP data was available for >88% of the area within MTBS perimeters. We will add this information for each fire into the new table proposed in comment #6.

6. Page 5, section 3.1: All of the numbers in this paragraph could go into a table and it might be easier to read.

R6: Thanks for the suggestion, we will add these into a new table.

7. Page 5, Line 7: It's mentioned here that other things besides fire may contribute
to NPP variability, but I don't think it was mentioned again. It's worth noting in the discussion whether climate or other factors might play a role in post-fire recovery of NPP.

R7: This is a good point – we will add some text in the discussion that addresses these factors.

Technical Corrections: 8. Page 3, Line 12: Some of the sites are not in the Northern Rocky Mountains.

R8: This will be re-worded in the revised manuscript.

9. Page 3, Lines 19-24: Pick past or present tense to be consistent throughout.

R9: This will be corrected in the revised manuscript.

10. Page 3, Line 26: "Canopy cover for each fire"- do you mean pre-fire canopy cover?

R10: This will be clarified in the revised manuscript.

Citation: Freeborn, P.H. M.J. Wooster, D.P. Roy, and M.A. Cochrane. 2014. Quantification of MODIS fire radiative power (FRP) measurement uncertainty for use in satellite based active fire characterization and biomass burning estimation. Geophysical Research Letters 41(6):1988-1994.

Response references: Hood SM, McHugh CW, Ryan KC, Reinhardt E, Smith SL (2007) Evaluation of a post-fire tree mortality model for western USA conifers. International Journal of Wildland Fire 16(6), 679–689. doi:10.1071/WF06122.

Holden, Z.A., Smith, A.M.S., Morgan, P., Rollins, M.G. and Gessler, P.E., 2005. Evaluation of novel thermally enhanced spectral indices for mapping fire perimeters and comparisons with fire atlas data. International Journal of Remote Sensing, 26(21), pp.4801-4808.

Ryan KC, Reinhardt ED (1988) Predicting post-fire mortality of seven western conifers.

BGD
Canadian Journal of Forest Research 18, 1291–1297. doi:10.1139/X88-199.

Smith, A.M., Talhelm, A.F., Johnson, D.M., Sparks, A.M., Kolden, C.A., Yedinak, K.M., Apostol, K.G., Tinkham, W.T., Abatzoglou, J.T., Lutz, J.A. and Davis, A.S., 2017. Effects of fire radiative energy density dose on Pinus contorta and Larix occidentalis seedling physiology and mortality. International Journal of Wildland Fire, 26(1), pp.82-94.

Sparks, A.M., Kolden, C.A., Talhelm, A.F., Smith, A., Apostol, K.G., Johnson, D.M. and Boschetti, L., 2016. Spectral indices accurately quantify changes in seedling physiology following fire: towards mechanistic assessments of post-fire carbon cycling. Remote Sensing, 8(7), p.572.

Sparks, A.M., Smith, A.M., Talhelm, A.F., Kolden, C.A., Yedinak, K.M. and Johnson, D.M., 2017. Impacts of fire radiative flux on mature Pinus ponderosa growth and vulnerability to secondary mortality agents. International Journal of Wildland Fire, 26(1), pp.95-106.

BGD

---

## Author Response (AR1)

Author's Response

Title: Fire intensity impacts on post-fire temperate coniferous forest net primary productivity
Author(s): Aaron M. Sparks et al.
MS No.: bg-2017-348
MS Type: Research article

Point-by-point response (in red text): pages 1-6
Marked-up manuscript version: pages 7-26
* * *
Anonymous Referee #1

Overall response: We would like to thank referee #1 for the supportive comments on the manuscript. We implemented their suggestions in our revised manuscript. Below we respond to each of the comments individually.

The manuscript by Sparks and colleagues examined how forest composition and fire intensity affected forest net primary productivity (NPP) following fire. The authors argue that higher fire intensity leads to progressively larger reduction in post-fire NPP among fire resistant and mixed-resistance communities, while fire intensity had little effect on the magnitude of NPP change in fire susceptible communities. The manuscript is well written and presents an analysis that provides novel insight into forest carbon dynamics following fire in a region where fire activity is likely to intensify over the coming century due to regional warming and drying. As detailed below, it seems there are several aspects of this analysis could be refined to further improve its rigor.

Primary comments
1. The manuscript states that, "Fire-affected pixels were grouped by FRP and FRE percentile classes (0-25, 25-50,: : :) for each fire" and then changes in post-fire NPP were evaluated among these percentile classes across all fires within a forest type. Why group pixels by fire-specific percentile class rather than by the absolute magnitude of fire intensity? Perhaps I am misunderstanding the approach, but let's say there are two fires of contrasting intensity, both of which occur in a fire-susceptible forest type. In this forest type, about 50% of pixels had FRE < 2000 MJ km-2 and about 50% of pixels had FRE between 2000 and 12000 MJ km-2 (figure 2). If the low-intensity fire only experienced FRE < 2000 MJ km-2 and the high intensity fire only experienced FRE > 2000 MJ km-2, then what happens when the pixels within each fire are grouped by the fire-specific percentile class and then these classes are subsequently grouped across fires? The 75-100th percentile class for the low severity fire might have FRE of, say, 1000-2000 MJ km-2, whereas this same percentile class for the high severity fire might encompass areas where the FRE was > 10,000 MJ km-2. You might expect a very different post-fire trajectory of NPP between these two fires for the same percentile class, but at present these would get grouped together, correct? This might somewhat explain why you don't see any difference between percentile classes in post-fire NPP trajectory for the fire-susceptible forest type.
R1: We thank the reviewer for this comment. These comments made us realize that we had worded the section poorly. As such, we have clarified this section in the revised version. To be clear, percentile classes were based on absolute magnitude of FRE (or FRP) by forest type, not for each individual fire.

2. The description of the statistical analysis is vague and the results do not present any statistics. Also, how do account for taking multiple pixels from the same fire and using them as independent samples, when in fact they are not independent?

R2: We thank the reviewer for their comment. We have included statistics relevant to the differences in NPP groups (see Figure 4). In terms of the comment regarding multiple pixels from the same fire, we believe that the text may have been unclear as we are using a census of fire-affected pixels and not a sample (i.e. all of the pixels within the fire perimeter are considered in the analysis, and not a sample of pixels within that perimeter).

Secondary comments
1. The researchers frequently note that there are dose-response relationships between fire intensity and post-fire changes in NPP. Given this focus, it would be worth including a figure that more explicitly shows this relationship. The figure could show the change in NPP one year after fire as a function of fire intensity for each of the three forest types.
R3: We think this is an excellent suggestion and have included a 1-year post-fire dose-response figure (Figure 3) in the revised manuscript.

2. Could it be that fire intensity is higher in fire-susceptible forests than mixed or fire-resistant forests not solely because of differences in trait characteristics, but rather because there is more biomass (fuel) in these forests? It could be worth normalizing fire intensity by forest biomass to see whether fire intensity per unit of fuel differs between these three broad forest types. The National Biomass and Carbon Data set 2000 (NBCD2000) could be a useful source of information for this endeavor (https://daac.ornl.gov/cgi-bin/dsviewer.pl?ds_id=1161)
R4: We thank the reviewer for this insight and suggestion. We downloaded this dataset and found that, on average, estimated biomass did not match trends in fire intensity (FRP, FRE). Average biomass per unit area (Mg ha$^{-1}$) decreased from mixed (157 Mg ha$^{-1}$) to fire-susceptible (129 Mg ha$^{-1}$) to fire resistant (110 Mg ha$^{-1}$), whereas fire intensity (FRP, FRE) decreased from fire-susceptible to mixed to fire resistant. We include this discussion point in an expanded results and discussion section (along with other potential drivers – e.g. climate, forest structural differences between the three types).

3. The manuscript includes figures showing the relative change in NPP following fire, but not the absolute change in NPP. It would be informative to show how the absolute magnitude of NPP changes after fire.
R5: We agree with this suggestion and have modified Figure 4 to show absolute change in NPP in addition to relative NPP.

4. Does including the FRP90th percentile add to the story? It seems somewhat redundant given the inclusion of FRPpeak and FRPmean.
R6: We agree with the reviewer and will remove 90$^{th}$ percentile FRP. Additionally, NPP response grouped by FRP and FRE were very similar, and to avoid redundancy, we have moved NPP response to peak FRP and mean NPP to supplemental files.

Line specific comments
1. Page 4, line 31: What does "Unburned pixels (nunburned = nFRP percentile group): : :" mean? Does this mean that you selected the same number of unburned pixels as there were pixels in the percentile group?
R7: That is correct – we have clarified this in the revised manuscript.

2. Page 5, lines 14-25: The researchers present the average and variation (presumably SD, but not defined) in fire intensity metrics for each forest type; however, Figure 2 shows that these metrics are very non-normally distributed. Consequently, mean and standard deviation are not appropriate summary statistics. The median and interquartile range would be more appropriate.
R8: We have corrected this in the revised manuscript.

3. Page 5, lines 29-30: the researchers state that, "in forests dominated by fire-resistant species, there was a stronger dose-response pattern for relative NPP grouped by FRE percentile class rather than FRP percentile class." This pattern is not particularly evident looking at figure 3. I would suggest providing additional evidence, or removing the statement.
R9: We have removed this statement.

4. Page 6, line 4: "The dose-response relationship was much weaker in forests dominated by fire susceptible species. There were few differences between percentile classes with only the highest FRE percentile class displaying lower relative NPP compared with other percentile classes." Is this based on a qualitative comparison, or statistical analysis?
R10: The revised text has clarified this in terms of the ANOVA analysis that was performed.

5. Page 6, line 14: The authors state that "generally, recovery trajectories [in NPP] were linear for all fire-resistant groups, except for a few fires were NPP begin decrease again around 2011." Looking at the supplemental figures, it appears that many, if not most, of the fires show non-linear changes in NPP after fire.
R11: This is a valid point. We have corrected the text to say that some fires had linear recovery trajectories, while most did not.

6. Page 7, lines 19-21: The authors note that the number of MODIS FRP observation differed between "fires with a clear dose- response relationship" and those with a "weak relationship." Does this suggest that there were differences in the number of MODIS FRP observations between forest types? Perhaps clarify what is meant by a clear relationship versus a weak relationship.
R12: The revised text clarifies this in terms of how dose-response studies are commonly analyzed and reported (e.g. Ruberg 1995). Specifically, the presence or absence of significant differences between fire intensity classes (and the unburned 'control' pixels) is emphasized in the revised text. Additionally, in response to reviewer #2 comment #1, we have added text in the discussion that addresses how the number of MODIS FRP observations affects FRP uncertainty and how it relates to the analyses that we performed.

7. Page 8, Conceptual framework: The following citations could bolster this section: i. Michaletz, S. T., E. Johnson, and M. Tyree. 2012. Moving beyond the cambium necrosis 1. Hypothesis of post-fire tree mortality: cavitation and deformation of xylem in forest fires. New Phytologist 194:254-263. ii. van Mantgem, P. J., J. C. Nesmith, M. Keifer, E. E. Knapp, A. Flint, and L. Flint. 2013. 1. Climatic stress increases forest fire severity across the western United States. Ecology Letters 16:1151-1156.
R13: Thanks for the suggestions, these were added.

8. Page 8, line 22: Always hesitant to say things are "obvious" in a paper.
R14: This was re-worded in the revised manuscript.

9. Supplemental figures: The Saddle fire appears to be missing the vertical line denoting the year in which the fire occurred. Also, what do the plotting characters and error represent in these figures? Mean and standard deviation?
R15: This was corrected and clarified.

Response references:
Ruberg, S.J., 1995. Dose response studies II. Analysis and interpretation. Journal of biopharmaceutical statistics, 5(1), pp.15-42.

Anonymous Referee #2

Overall response: We would like to thank referee #2 for the supportive comments on the manuscript. We have implemented their suggestions in our revised manuscript. Below we respond to each of the comments individually.

This is an interesting paper detailing how NPP varies with fire severity across 15 large fires in the western U.S. MODIS satellite data at the 1-km pixel scale was used, giving a coarse view of fire severity effects on productivity. The paper addresses relevant scientific questions, presents novel results, and reaches substantial conclusions. However, some aspects of the paper, both major and minor, could be improved. General and specific comments follow.

General comments:
1. Freeborn et al. 2014 reported that differences in per-pixel FRP measured near simultaneously have a standard deviation of 27%, and that clumping pixels helps a lot (50-pixel aggregation reduces uncertainty to 5%; citation at end of comments). This seems like a relevant issue for the current study, since it uses pixel-level data. Would including the uncertainty in the analysis change the results or the interpretation of the results?
R1: This is a great point. Spatiotemporal aggregations of observations have been shown to reduce uncertainties in sums of FRP. Due to the pixel level variability in ages and species composition in the forests analyzed in this paper, only temporal FRP aggregations were employed (i.e. FRE, mean FRP). It seems clear that further aggregation through time, if data were available, would reduce the uncertainty in FRP aggregations and improve the strength of our reported relationship (which is in line with Freeborn et al. (2014) for temporal aggregation). Given that our method reports changes in pixel-level NPP that are varying as functions of structure, age and composition to a fire, we do not think that spatially aggregating the FRP observations would be appropriate. Freeborn et al (2014) advocate this methodology for abiotic relationships between FRP and emissions (fuel is fuel regardless of location), however, we are relating FRP to biotic responses of living organisms that are clearly related to the fire resistance/sensitivity of the species.

2. I don't find the conceptual framework (page 8 and Figure 4) to be very strong. The authors state that they are linking individual tree-level processes to fire intensity and forest growth and productivity. But they go on to say in the Limitations section that understory vegetation may recover rapidly and make it appear that the overstory recovers rapidly. It doesn't seem that the authors can actually say much about individual tree mortality, given the heterogeneity of fires on the ground, the large size of the pixels being used, and the lack of on-the-ground severity measurements. Couldn't it be that shrubs are what are responding post-fire rather than trees?
R2: Sparks et al. 2016 and Smith et al. 2017 observed mechanistic links between FRP and sapling mortality and productivity. These, and other studies (e.g. Sparks et al. 2017), also collectively demonstrated that the mechanism scaled from the saplings in a laboratory fire to mature trees in stand-scale fires. Prior studies (Ryan and Reinhardt 1988; Hood et al. 2007) had previously reported similar relationships between proxies of fire intensity and mature tree mortality. Although the current paper suggests that this may further scale from the watershed to the regional scale, we agree that this is not yet proven. As such, we have adjusted the text to be more circumspect and cautious of a regional scale relationship.

3. Finally, I agree with the first reviewer in questioning why the authors grouped the FRP and FRE into percentile classes, because then it's difficult to compare actual FRP and FRE in terms of their effect on NPP across fires- you've limited the analysis to within fire differences. Similarly, I also question why

relative NPP rather than absolute NPP is shown in the supplemental figures. Are there are interesting absolute differences among forest types?

R3: This section was poorly worded and was clarified in the revised text. Percentile classes were based on absolute FRP/FRE magnitude by forest type, not for each individual fire. Per reviewer #1 comment #3 we have included absolute NPP in Figure 4.

Specific comments:

4. Page 4, Line 1: MTBS only includes fires 1000 acres and bigger: are the authors able to verify through other data sources that these areas haven't burned since 1984? Does it matter?

R4: This is a great point. We mapped smaller burned areas within each MTBS polygon using the Normalized Burn Ratio Thermal Index (Holden et al. 2005) computed by Google Climate Engine (climateengine.org) annually from 1984 to the present. Google Climate Engine uses data from Landsat 4, 5, 7, and 8 depending on availability and cloud cover to produce 30 m spatial resolution datasets. Using these data we found that, on average, less than 1.5% of the MTBS polygon area burned between 1984 and the year that each fire burned. We have included this information for each fire in Table 1.

5. Page 4, MODIS datasets: Was FRP available for all pixels inside the MTBS perimeters?

R5: On average, FRP data was available for >88% of the area within MTBS perimeters. We have added this information for each fire into Table 1.

6. Page 5, section 3.1: All of the numbers in this paragraph could go into a table and it might be easier to read.

R6: Thanks for the suggestion. We have removed the numbers for easier reading, and refer readers to Figure 2.

7. Page 5, Line 7: It's mentioned here that other things besides fire may contribute to NPP variability, but I don't think it was mentioned again. It's worth noting in the discussion whether climate or other factors might play a role in post-fire recovery of NPP.

R7: This is a good point – we have added text in the discussion that addresses these factors.

Technical Corrections:

8. Page 3, Line 12: Some of the sites are not in the Northern Rocky Mountains.

R8: This was re-worded in the revised manuscript.

9. Page 3, Lines 19-24: Pick past or present tense to be consistent throughout.

R9: This was corrected in the revised manuscript.

10. Page 3, Line 26: "Canopy cover for each fire"- do you mean pre-fire canopy cover?

R10: This was clarified in the revised manuscript.

Citation: Freeborn, P.H. M.J. Wooster, D.P. Roy, and M.A. Cochrane. 2014. Quantification of MODIS fire radiative power (FRP) measurement uncertainty for use in satellite based active fire characterization and biomass burning estimation. Geophysical Research Letters 41(6):1988-1994.

Response references:

Hood SM, McHugh CW, Ryan KC, Reinhardt E, Smith SL (2007) Evaluation of a post-fire tree mortality model for western USA conifers. International Journal of Wildland Fire 16(6), 679–689. doi:10.1071/ WF06122.

Holden, Z.A., Smith, A.M.S., Morgan, P., Rollins, M.G. and Gessler, P.E., 2005. Evaluation of novel thermally enhanced spectral indices for mapping fire perimeters and comparisons with fire atlas data. International Journal of Remote Sensing, 26(21), pp.4801-4808.

Ryan KC, Reinhardt ED (1988) Predicting post-fire mortality of seven western conifers. Canadian Journal of Forest Research 18, 1291–1297. doi:10.1139/X88-199.

Smith, A.M., Talhelm, A.F., Johnson, D.M., Sparks, A.M., Kolden, C.A., Yedinak, K.M., Apostol, K.G., Tinkham, W.T., Abatzoglou, J.T., Lutz, J.A. and Davis, A.S., 2017. Effects of fire radiative energy density dose on Pinus contorta and Larix occidentalis seedling physiology and mortality. International Journal of Wildland Fire, 26(1), pp.82-94.

Sparks, A.M., Kolden, C.A., Talhelm, A.F., Smith, A., Apostol, K.G., Johnson, D.M. 
[revised manuscript text omitted]
 | 18,778 | *Pseudotsuga menziesii*PSME, *Pinus ponderosa*PIPO, *Larix occidentalis*LAOC, *Picea engelmannii*PIEN, *Abies lasiocarpa*ABLA, *Pinus albicaulis*PIAL | June 28, 2007 | 0 | 86.2 |
| Arnica | 4,556 | *Pinus contorta*PICO | September 23, 2009 | 0 | 76.9 |
| Bridge | 15,116 | *Picea engelmannii*PIEN, *Abies lasiocarpa*ABLA | July 18, 2007 | 8.9 | 91.0 |
| Columbine | 7,115 | *Picea engelmannii*PIEN, *Abies lasiocarpa*ABLA, *Pinus albicaulis*PIAL | August 9, 2007 | 1.6 | 91.2 |
| East | 7,145 | *Picea engelmannii*PIEN, *Abies lasiocarpa*ABLA, *Pinus albicaulis*PIAL, *Pinus contorta*PICO | August 8, 2003 | 0.6 | 93.6 |
| Fawn Peak | 31,870 | *Pseudotsuga menziesii*PSME, *Pinus ponderosa*PIPO, *Larix occidentalis*LAOC, *Picea engelmannii*PIEN, *Abies lasiocarpa*ABLA, *Pinus albicaulis*PIAL | June 30, 2003 | 0.3 | 91.8 |
| Fool Creek | 22,186 | *Picea engelmannii*PIEN, *Abies lasiocarpa*ABLA, *Pinus albicaulis*PIAL | June 28, 2007 | 2.0 | 89.9 |
| Little Salmon | 13,598 | *Pseudotsuga menziesii*PSME, *Pinus ponderosa*PIPO, *Larix occidentalis*LAOC, *Picea engelmannii*PIEN, *Abies lasiocarpa*ABLA | July 18, 2003 | 0.1 | 78.2 |
| Meriwether | 7,762 | *Pseudotsuga menziesii*PSME, *Pinus ponderosa*PIPO, *Larix occidentalis*LAOC | July 21, 2007 | 0.2 | 98.1 |
| North Fork | 6,774 | *Pseudotsuga menziesii*PSME, *Pinus ponderosa*PIPO | August 1, 2009 | 1.1 | 92.1 |
| Saddle | 12,706 | *Pseudotsuga menziesii*PSME, *Pinus ponderosa*PIPO, *Larix occidentalis*LAOC, *Picea engelmannii*PIEN, *Abies lasiocarpa*ABLA | August 18, 2011 | 0.2 | 80.1 |
| Sawmill | 6,015 | *Pseudotsuga menziesii*PSME, *Pinus ponderosa*PIPO, *Larix occidentalis*LAOC | July 13, 2007 | 3.0 | 95.5 |
| Shower Bath | 19,911 | *Pseudotsuga menziesii*PSME, *Pinus ponderosa*PIPO | July 17, 2007 | 1.1 | 78.0 |
| South Fork | 11,494 | *Pseudotsuga menziesii*PSME, *Pinus ponderosa*PIPO | August 7, 2006 | 1.2 | 82.3 |
| Tatoosh | 20,185 | *Pseudotsuga menziesii*PSME, *Pinus ponderosa*PIPO, *Larix occidentalis*LAOC, *Picea engelmannii*PIEN, *Abies lasiocarpa*ABLA, *Pinus albicaulis*PIAL | August 22, 2006 | 2.3 | 92.4 |

[a]Conifer species codes: ABLA – *Abies lasiocarpa*, LAOC – *Larix occidentalis*, PIAL – *Pinus albicaulis*, PICO – *Pinus contorta*, PIEN – *Picea engelmannii*, PIPO – *Pinus ponderosa*, PSME – *Pseudotsuga menziesii.*

**Figure 2.** Location of study fires overlaid on current distribution of U.S. forest types classified using relative fire resistance information in the literature and the LANDFIRE Existing Vegetation Type (EVT) 30-m product.

[Figure]

**Figure 2.** Fire radiative power (a) and fire radiative energy (b) distributional statistics grouped by dominant forest composition (fire-resistant to fire-susceptible). Black arrows indicate mean values.

[Figure]

**Figure 3.** Fire intensity impacts on one-year post-fire NPP observed in forests dominated by species varying from fire-resistant to fire susceptible (first column – third column). Distributional statistics are shown for NPP grouped by: a) peak FRP percentile class and b) FRE percentile class.

[Figure]

**Figure 4.** Fire intensity impacts on net primary productivity.  a) FRE dose impacts on absolute NPP (g C m$^{-2}$ yr$^{-1}$) and relative NPP (%)  observed in forests dominated by species varying from fire-resistant to fire-susceptible (first column – third column).  NPP is grouped by  FRE percentile classes and shading represents 95% confidence intervals in all panes. Grey dotted line marks fire year. b) Results from ANOVA with a post hoc Tukey's honest significant difference test (α = 0.05). Black bars indicate years where relative NPP groups differed.

[Figure]

**Figure 5.** Conceptual framework for quantifying impacts of fire intensity on physiology, growth, and vulnerability of coniferous forests.

[Figure]